# Risky Alcohol Consumption in the Elderly: Screening and Brief Intervention from Primary Care. The ALANE Study, a Randomized Clinical Trial

Pere Torán [1,2,3,4,†], Susanna Montesinos [1,†], Alba Pachón-Camacho [1], Galadriel Diez-Fadrique [1], Irene Ruiz-Rojano [1,5], Ingrid Arteaga [1,5], Guillem Pera [1], Núria Montellà-Jordana [1,3], Pilar Montero-Alía [1,6], Carmina Rodríguez-Pérez [1,6], Llorenç Caballeria [1,5] and Carla Chacón [1,5,7,*]

1  Research Support Unit of the North Metropolitan Area (USR Metro-Nord), University Institute for Primary Care Research Jordi Gol i Gurina (IDIAP Jordi Gol), 08303 Barcelona, Spain; ptoran.bnm.ics@gencat.cat (P.T.); smontesinos.bnm.ics@gencat.cat (S.M.); apachonca.mn.ics@gencat.cat (A.P.-C.); galadrieldiez21@gmail.com (G.D.-F.); ireneruiz.mn.ics@gencat.cat (I.R.-R.); iarteaga@gencat.cat (I.A.); gpera@idiapjgol.info (G.P.); nmontella.bnm.ics@gencat.cat (N.M.-J.); pmonteroalia@gmail.com (P.M.-A.); crodriguezp.bnm.ics@gencat.cat (C.R.-P.); lcaballeriarovira@gmail.com (L.C.)
2  Directorate of Primary Health Care of the North Metropolitan Area, Catalan Institute of Health (ICS), 08916 Badalona, Spain
3  Multidisciplinary Research Group in Health and Society (GREMSAS) (2021-SGR-0148), University Institute for Primary Care Research Jordi Gol i Gurina (IDIAP Jordi Gol), 08007 Barcelona, Spain
4  Department of Medicine, Faculty of Medicine, Universitat de Girona, 17004 Girona, Spain
5  Research Group on Liver Diseases in Primary Health Care (GRemHAp), University Institute for Primary Care Research Jordi Gol i Gurina (IDIAP Jordi Gol), 08007 Barcelona, Spain
6  Research Group on Interventions and Determinants Associated with Healthy Ageing (GRIADES), University Institute for Primary Care Research Jordi Gol i Gurina (IDIAP Jordi Gol), 08007 Barcelona, Spain
7  Sant Joan de Déu Teaching Campus, University of Barcelona, 08034 Barcelona, Spain
*  Correspondence: cchaconv.mn.ics@gencat.cat; Tel.: +34-93-741-53-38
†  These authors contributed equally to this work and share first authorship.

**Abstract:** Background: Risky alcohol consumption (RAC) can lead to alcohol-related liver disease (ALD). Liver cirrhosis caused by ALD continues to increase as alcohol consumption continues unabated. In turn, the elderly are more sensitive to alcohol. Population ageing calls for preventive activities to improve their health. Brief interventions have proven to be cost-effective in addressing risk behaviours. Aim: We aimed to analyse the prevalence of RAC in people > 64 years and to assess the effect of a brief intervention in the subgroup of risky consumers. Methods: population-based study in two phases: (1) Phase I: Cross-sectional, descriptive multicentre study of prevalence of RAC in people > 64 years. (2) Phase II: Cluster randomized, controlled, single-blind, community-based clinical trial with two comparison groups of subjects with RAC, to assess the effectiveness of a brief intervention compared to standard practice in reducing alcohol consumption in primary care. Results: Out of the 932 subjects, 455 (49%) (268 men (64%) and 187 women (36%)) had an alcohol consumption that was considered to be risky. Overall, the brief intervention was effective in reducing alcohol consumption showing 1.8 OR ($p = 0.030$). That effect was caused by women whose group showed 3.3 OR ($p = 0.009$). There was no effect on men ($p = 0.468$). Conclusions: RAC in the elderly is very high, far more in men than in women. A brief intervention was successful in reducing alcohol consumption but not below risk levels. Further research is needed to determine which types of interventions are most effective in this population subgroup.

**Keywords:** risky alcohol consumption; elderly; brief intervention; alcohol-related liver disease; primary care

## 1. Introduction

Alcohol consumption is the second most significant health risk factor after tobacco use, being three times more important than diabetes and five times more important than asthma [1]. Risky alcohol consumption is defined as consumption that over time increases the probability that an individual will suffer adverse health outcomes [2]. Excessive alcohol consumption can cause or exacerbate up to 60 different diseases and is also a determinant factor in domestic and gender-based violence incidents, work-related accidents and traffic accidents. Alcohol-related deaths account for 9.3% of the potential years of life lost: 70% are due to acute conditions, the main cause being unintentional accidents, mainly traffic accidents [1]. Most recent data indicate that 3.8% of global deaths are attributed to alcohol (6.3% in men; 1.1% in women) [3].

Risky alcohol consumption is directly linked to the development of alcohol-related liver disease (ALD). ALD is a huge global health challenge as risky alcohol consumption continues unabated [4]. It has alarmingly high prevalences in various countries and, in some, its incidence is also on the rise. Epidemiological data reveal that cirrhosis and liver cancer, the final stage of liver diseases, stand as major causes of mortality rates worldwide and place a serious economic burden on healthcare systems, exacerbated by factors such as unemployment and a notable decline in overall quality of life [5–9]. The natural course of ALD is characterized by the gradual development of liver fibrosis due to persistent inflammation stemming from the harmful effects of alcohol on liver cells. This progression can ultimately lead to cirrhosis in a significant portion of affected individuals. Key risk factors for cirrhosis include the duration and intensity of alcohol consumption, as well as the presence of other factors contributing to chronic liver disease, particularly hepatitis B or C infections and metabolic syndrome [4,10]. In general, patients are not diagnosed or given any specific intervention during the developmental stage of fibrosis as they have no symptoms [11]. Thus, detecting ALD hinges on identifying individuals with risky alcohol consumption patterns and being able to intervene on their behavioural patterns in order to redirect them at the earliest possible stage.

In particular, the elderly are more sensitive to alcohol than younger adults, a sensitivity that is enhanced by increased consumption of over-the-counter and prescription drugs. Furthermore, with age, body mass relative to total fat volume decreases, leading to an increase in the overall distribution of alcohol; liver enzymes that metabolize alcohol become less efficient and the sensitivity of the central nervous system increases [12]. There is scarce information on the prevalence of risky alcohol consumption in the elderly. However, studies highlight the potential interaction between medication and alcohol especially among this population [12]. To date, reports carried out including the population over 65 years of age set the alcohol levels referred to for the young and adult population as limits and do not take into account factors such as polymedication. Therefore, we still do not have any references on the real prevalence of risk drinkers among the elderly [13].

Brief interventions have been shown in numerous national and international studies to be effective and cost-effective [14]. The World Health Organisation (WHO) has estimated that a brief intervention from primary care with a coverage rate of 25% would prevent 91 years of illness and premature death per 100,000 people, 9% of all those caused by alcohol. The cost-effectiveness of brief intervention is 1969 euros per disability-free life-year averted; the second most cost-effective health intervention, only behind tobacco use intervention [15]. It is generally accepted that single interventions of about 10 min duration, reinforced with written material (leaflets), can reduce alcohol intake by 35% and bring 45–50% of patients below risk levels. However, most studies along these lines exclude older patients and, moreover, are carried out exclusively in males [16,17]. Currently recommended brief interventions are based on the 5As model for brief interventions proposed by the United States Preventive Services Task Force (USPSTF) [18] and are reflected in the Preventive Activities and Health Promotion Programme (PAPPS) of the semFYC (Spanish Society of Family and Community Medicine) as: assess advice, agree, assist and arrange [19].

Considering the high vulnerability of the elderly population to the effects of alcohol, the cost-effectiveness of brief interventions in risky drinkers and the lack of information available in this age group, we aimed to analyse the prevalence of risky alcohol consumers in people over 64 years of age and to assess the effect of a brief intervention in the subgroup of risky consumers.

## 2. Materials and Methods

### 2.1. Study Design and Population

Population-based study in 2 phases: (1) Phase I: Cross-sectional, descriptive multicentre study of prevalence of risky alcohol consumption (RAC) in people > 64 years of age. (2) Phase II: cluster-randomized (by healthcare centre), controlled, single-blind, community-based clinical trial with 2 comparison groups of subjects with RAC, to assess the effectiveness of a brief intervention compared to standard practice in reducing alcohol consumption in primary care.

The study was designed including individuals ascribed to 25 Primary Healthcare Centers of the area of Barcelonès Nord and Maresme (Catalonia, Spain), which covers a population of 700,000 inhabitants, of which 82,903 are aged >64 years old.

The sample was randomly selected from the database of the Primary Care Information System (SIAP) which includes all individuals with national healthcare cards and is equivalent to the population census of Catalonia. This database includes all the individuals ascribed to a Primary Healthcare Centre of the zone, regardless of whether they have been attended or not.

Inclusion criteria: (1) Phase I: Population of both sexes aged >64 years old ascribed to the participating Primary Healthcare Centers, who voluntarily provided written informed consent and accepted to participate in the study. (2) Phase II: Subjects from Phase I who met criteria for RAC (these criteria are detailed below in the methodology).

Exclusion criteria: (1) Phases I and II: Subjects with conditions making data collection and follow-up difficult, such as incapacitating conditions, cognitive impairment or individuals in long-term care facilities. (2) Phase II: Subjects meeting criteria for alcohol dependence according to the International Classification of Diseases (ICD-10) [20].

### 2.2. Sample Size

An alpha risk of 0.05 was determined for a precision of $+/- 3\%$ and an estimated overall sex prevalence of risk drinkers in >64 years of age of 30%. (1) Phase I: A random sample of 1000 subjects aged >64 years was estimated, assuming a 25% loss rate (10% of wrong telephone numbers, 10% of patients who did not want to participate and 5% who accepted but did not show up for the visit); 1333 individuals aged >64 years needed to be invited to participate. (2) Phase II: It was estimated that 30% of the 1000 patients would be risky drinkers (300 subjects for Phase II). Accepting an alpha risk of 5% and a beta risk of 20% with bilateral contrasts, 150 risky drinkers in the intervention group (brief intervention) and 150 risky drinkers in the control group (minimal advice) were needed, with the aim of detecting a difference between the two groups equal to or greater than 15% of patients who stopped being risky drinkers. A loss rate of 20% was taken into account (2% failure, 1% institutionalization, 2% transfer, and 15% dropout).

### 2.3. Study Variables

#### 2.3.1. Sociodemographic Variables

Age, sex, birth country, years of residence in Spain, ethnicity, educational level, and marital status were considered. Primary care centre was used as unit of randomization for the 2nd phase.

#### 2.3.2. Variables Related to Alcohol Consumption

(a)　Primary variable. Alcohol consumption was recorded as standard drink units (SDU). Monday to Friday consumption was differentiated from weekend and occasional

consumption; and total weekly and last month consumption was quantified. The type of alcoholic beverage consumed was also collected. Considering that one SDU is equivalent to 10 g of pure alcohol, a risky consumer was defined as follows:

- Males: Consumption of more than 1 SDU per day; or more than 7 SDU per week; or more than 2 SDU on one drinking occasion; or any consumption if they had any pathology made worse by alcohol consumption or were taking drugs that interacted with alcohol (according to PAPPS criteria) [2,19].
- Females: From 1 SDU per day; or 7 SDU per week; or more than 1 SDU on one drinking occasion; or any consumption if they had a pathology made worse by alcohol consumption or were taking drugs that interacted with alcohol (according to PAPPS criteria) [2,19].

(b)  Outcome variables. Three scenarios were considered:

- Success: Cases where the number of SDU was reduced below the levels considered to be at risk.
- Partial success: Cases where the consumption of SDU decreased, but did not fall below the risk threshold.
- Failure: Cases where the consumption of SDU was not reduced or even increased.

### 2.3.3. Other Variables/Outcomes

(a)  Clinical profile: Number of times visiting their doctor in the last year, tobacco use, level of dependence for daily living activities according to Barthel test [21], level of family dysfunction according to family Apgar test [22], pathological history and pharmacological treatments (number of drugs if taken and type of drugs that interact with alcohol) recorded in computerized medical history or self-reported, inclusion in the Domiciliary Care programme, body mass index, nutritional status according to the Mininutritional Assessment (reduced MNA) [23], cognitive impairment according to the Lobo mini-cognitive examination [24], depression and anxiety according to the Goldberg test [25], risk of falls according to the Timed up and go test [26] and insomnia according to the COS test [27].

(b)  Analytical parameters: Blood count and biochemistry, including glucose, transaminases, bilirubin, total cholesterol, triglycerides, creatinine, total protein, albumin, sodium and potassium.

### *2.4. Data Collection*

The necessary subjects were recruited from a random selection obtained from the SIAP by telephone contact carried out by specific staff through a call centre. Those who agreed to participate were invited to a first visit (visit 1), at the health centre to which the subject belonged, to detect risk drinkers. If the first contact was null, up to a maximum of 6 more calls were made on different days and times. If the subject was unable to attend the primary care centre, the visit was conducted at their home. During this visit, the selection criteria were reviewed, an information sheet about the study was handed out and the informed consent form was asked to be signed. Subjects underwent a blood test. Risky consumers were identified using the computerized clinical history and a self-drafted questionnaire on healthy lifestyle habits that included: (1) questions on quantity and frequency of habitual alcohol consumption in the last 3 months, and (2) drug consumption. Criteria for alcohol dependence were ruled out. In case of being classified as a risky drinker, the rest of the variables listed above were collected. Subjects classified as risky drinkers entered Phase II of the study. They were included in the intervention group or in the control group depending on the health centre to which they belonged (patients excluded because they met the criteria for alcohol dependence continued with the follow-up plan established by their Basic Health Unit). Health centres were divided into "intervention" (13 centres) or "control" (12 centres), randomly.

### 2.4.1. Intervention Group

In the second part of visit 1, subjects belonging to the "intervention centers" received the brief intervention standardized according to the recommendations of the PAPPS (Spanish Society of Family and Community Medicine) [19]. The stage of change was explored in order to adapt the brief intervention to the patient's attitude at each moment. In order to implement this intervention, the healthcare professionals who carried it out received specific training to standardize the procedure. A total of 3 brief interventions were performed: (1) at the beginning (visit 1), (2) after one month (visit 2), and (3) after six months (visit 3). A final visit (visit 4) was made to both the intervention and control groups, one year after the start of the project.

### 2.4.2. Control Group

Subjects belonging to the "control centers" received, at visit 1, the minimum advice defined as that given in the usual consultation and which was standardised according to the PAPPS recommendations; they were not seen again until the last visit. At this last visit, which took place one year after assignment to each group (visit 2 for the control group and visit 4 for the intervention group), a follow-up was carried out in which alcohol consumption was again quantified, the study tests were passed and a blood test was performed again.

### 2.5. Ethics and Confidentiality

The protocol was approved on 30 June 2010 by the Ethics Committee of the Fundació Gol i Gurina (P10/35) (Barcelona, Spain) which followed the Declaration of Helsinki. All subjects provided written informed consent before inclusion.

### 2.6. Data Analysis

A univariate descriptive analysis was performed for all variables collected (mean and standard deviation for symmetric quantitative variables, median and interquartile range for skewed quantitative variables and frequency and percentage for qualitative variables), and, in Phase I, the prevalence of risk drinkers and its 95% confidence interval were used as a measure of frequency. This prevalence was estimated overall and by sex.

To analyse the effectiveness of the brief intervention we used the chi-square test between intervention outcome (success/partial success/failure) and group (control/intervention). Chi-square was used to compare two qualitative variables (Fisher's exact test when the expected value of any cell was less than 5) and *t*-test to compare one continuous variable between 2 groups.

In Phase 2, multivariate mixed effects logistic models were used to assess the effect of the intervention in reducing the alcohol intake (success vs. partial/failure and success/partial vs. failure as outcomes). Random intercepts for the primary care centres and random slopes for the intervention were included. Potential confounders were included as fixed effects only if they were statistically different between the control and intervention groups. Overall and by sex results are provided. Both analyses, intention to treat and per protocol, were performed.

All the statistical tests were performed bilaterally with a significance of 5%. The analyses were performed with the Stata v18 statistical package.

## 3. Results

Data collection for the study was conducted between 2012 and 2015. The population aged >64 years assigned to participating primary care centres included 82,903 inhabitants. Of these, 1060 individuals were randomly selected to be initially invited to participate in screening for risky alcohol consumption in the elderly.

A total of 932 (88%) individuals agreed to participate and their data were finally analyzed in Phase I. An amount of 455 (49%) were classified as risk drinkers and, so, included in Phase II of the study. An amount of 213 subjects were included in the control

group and 242 in the intervention group. An amount of 137 subjects (64%) completed the final visit in the control group, while, in the intervention group, 113 (47%) did so. It should be noted that 167 subjects (69%) of the intervention group completed the entire intervention but only 88 managed to complete the entire intervention along with the final visit. Therefore, a total of 225 subjects (137 in the control group and 88 in the intervention group) were analysed following a per protocol analysis (Figure 1), and 250 following an intention-to-treat analysis.

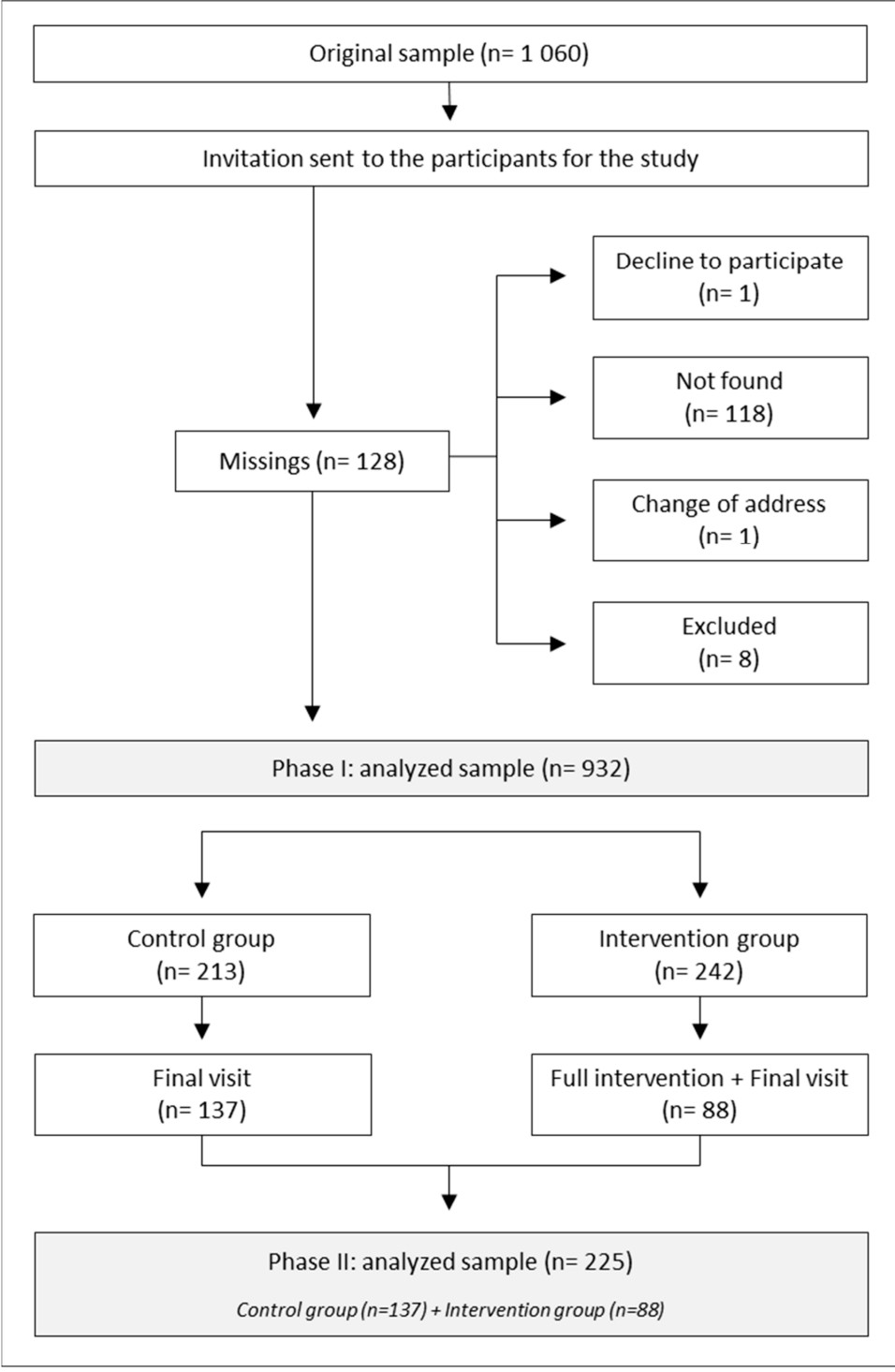

**Figure 1.** Sample flow diagram.

### 3.1. Characteristics of the Sample

Of the sample analysed, 419 individuals were men (45%) and 513 women (55%), with a mean age of 71 years (SD ± 5 years) (range: 64–93 years). Table 1 shows the main descriptive characteristics and affiliation variables of the study sample. Among the sample, 99% were Caucasians from Spanish origin.

**Table 1.** Baseline characteristics and affiliation variables of the 932 subjects included in the study.

| Sex | n | % |
| --- | --- | --- |
| Men | 419 | 45% |
| Women | 513 | 55% |
| **Age (mean/SD)** | 71 | 5 |
| <70 | 472 | 51% |
| 70–75 | 263 | 28% |
| >75 | 197 | 21% |
| **Educational level** | | |
| Illiterate | 37 | 4% |
| Can read/write | 214 | 23% |
| Elementary school | 494 | 53% |
| Secondary school | 121 | 13% |
| University | 66 | 7% |
| **Consumption of drugs** | 681 | 73% |
| NSAID | 157 | 17% |
| Painkillers | 297 | 32% |
| Nitrates | 26 | 3% |
| Dicoumarinics | 46 | 5% |
| Antidepressants | 100 | 11% |
| Anxiolytics, antiepilectics and hypnotics | 212 | 23% |
| Antihistamines antiallergics | 12 | 1% |
| Antipsychotics | 12 | 1% |
| Lithium carbonate | 0 | 0% |
| Antiparkinsonian | 3 | 0% |
| Anti H1/antiemetics | 19 | 2% |
| Morphine and opioids | 17 | 2% |
| Methotrexate | 2 | 0% |
| Antidiabetics | 141 | 18% |
| AntiHTA/alpha blockers | 474 | 51% |

### 3.2. Prevalence of Alcohol Consumption

Out of the 932 individuals, 455 (49%) (268 men (64%) and 187 women (36%)) had an alcohol consumption that was considered to be risky. Notably, 63% of the sample consumed alcohol while only 14% consumed alcohol below the risk thresholds (Table 2).

**Table 2.** Prevalence of alcohol consumption and categorization by risk status.

| | Men | | Women | | Total | |
| --- | --- | --- | --- | --- | --- | --- |
| | n | % | n | % | n | % |
| Abstainer | 88 | 21% | 261 | 51% | 349 | 37% |
| Non-risk drinker | 62 | 15% | 65 | 13% | 127 | 14% |
| Risk drinker | 268 | 64% | 187 | 36% | 455 | 49% |
| Alcohol dependence syndrome | 1 | 0.2% | 0 | 0% | 1 | 0.1% |

The percentage of alcohol dependence syndrome is counted and included in the total percentage of the group of risk drinkers.

The prevalence of risk drinkers was very high, much higher in men than in women. It should be noted that only 36% of men and 5% of women had a regular high consumption

(>7 SDU/week), with a 24% of them being risk drinkers because they combined sporadic or low consumption of alcohol with medication (Tables 3 and 4).

**Table 3.** Prevalence of risky alcohol consumption by sex.

|  | Percentage | CI95% | |
|---|---|---|---|
| Men | 64.0% | 59.2% | 68.6% |
| Women | 36.5% | 32.3% | 40.8% |
| Total | 48.8% | 45.6% | 52.1% |

$p < 0.001$ for a chi-squared test comparing prevalence of risky alcohol consumption between men and women.

**Table 4.** Prevalence of alcohol consumption by risk categories.

|  | Men | | Women | | Total | |
|---|---|---|---|---|---|---|
|  | **n** | **%** | **n** | **%** | **n** | **%** |
| **Abstainer** | 88 | 21% | 261 | 51% | 349 | 37% |
| **Non-risk drinker** | | | | | | |
| Sporadic consumption | 23 | 5% | 40 | 8% | 63 | 7% |
| Regular non-risk consumption risk | 39 | 9% | 25 | 5% | 64 | 7% |
| **Risk drinker** | | | | | | |
| Consumption sporadic + medication | 21 | 5% | 36 | 7% | 57 | 6% |
| Low consumption + medication | 80 | 19% | 87 | 17% | 167 | 18% |
| High consumption on one occasion | 6 | 1% | 2 | 0.4% | 8 | 1% |
| High consumption on one occasion + medication | 8 | 2% | 38 | 7% | 46 | 5% |
| High regular consumption | 31 | 7% | 6 | 1% | 37 | 4% |
| High regular consumption + medication | 122 | 29% | 18 | 4% | 140 | 15% |
| Alcohol dependence | 1 | 0.2% | 0 | 0% | 1 | 0% |

$p < 0.001$ for a Fisher's exact test comparing the distribution of alcohol consumption between men and women. Sporadic consumption < 1 SDU/week; regular non-risk consumption, 1–7 SDU/week; high consumption, on one occasion > 2 SDU all at one (men)/>1 SDU all at once (women); high regular consumption, >7 SDU/week.

Table 5 shows a comparison of the affiliation variables with respect to alcohol consumption. Men had a higher prevalence of risky alcohol consumption (64%) and women had the highest percentage of abstainers (51%). It stands out that the lowest prevalence of risk consumption among age groups was in the >75 years. We also found a trend with education, increasing risky consumption with education. This result was similar by sex, but it was more evident among those older than 70.

**Table 5.** Comparison of affiliation variables with respect to alcohol consumption (n = 931).

|  | Abstainer | | Non-Risk Drinker | | Risk Drinker | | |
|---|---|---|---|---|---|---|---|
| **Sex** | **n** | **%** | **n** | **%** | **n** | **%** | **p** |
| Men | 88 | 21% | 62 | 15% | 268 | 64% | <0.001 |
| Women | 261 | 51% | 65 | 13% | 187 | 36% | |
| **Age (mean SD)** | 71.6 | 5.0 | 71.5 | 5.0 | 70.8 | 4.2 | 0.038 |
| <70 | 176 | 37% | 62 | 13% | 233 | 49% | 0.003 |
| 70–75 | 82 | 31% | 33 | 13% | 148 | 56% | |
| >75 | 91 | 46% | 32 | 16% | 74 | 38% | |
| **Educational Level** | | | | | | | <0.001 |
| Illiterate | 24 | 65% | 2 | 5% | 11 | 30% | |
| Can read/write | 103 | 49% | 31 | 15% | 77 | 36% | |
| Elementary school | 172 | 36% | 71 | 15% | 238 | 49% | |
| Secondary school | 34 | 28% | 15 | 12% | 74 | 60% | |
| University | 14 | 23% | 4 | 6% | 44 | 71% | |

One subject with alcohol dependence not included.

Patterns of Alcohol Consumption among Drinkers

Table 6 summarizes the weekly consumption patterns, excluding abstainers. In this table, we observe that men had a much higher consumption than women, even when we take into account if they were risk or non-risk drinkers.

**Table 6.** Patterns of alcohol consumption per SDU a week (n = 583).

|  | Men | | Women | | Total | | |
|---|---|---|---|---|---|---|---|
|  | **Median** | **IQR** | **Median** | **IQR** | **Median** | **IQR** | **Range** |
| Non-risk drinker | 2 | 6 | 0 | 2 | 1 | 3 | 0–7 |
| Risk drinker | 9 | 10 | 2 | 6 | 7 | 12 | 0–43 |
| Alcohol dependence | 35 | 0 | - | - | 35 | 0 | 35–35 |

Maximum SDU per drinking occasion: 8.

The highest consumption of alcohol among risk drinkers is observed at lunch and dinner, especially of wine. On each occasion of drinking, the most common intake was 1 SDU per occasion, being 1 SDU on all occasions with >1% of consumption. Alcohol consumption increased slightly during the weekends. Beer was the next most consumed alcoholic beverage, followed by liquor and fortified wine. The pattern of consumption was as follows in descending order: lunch, dinner and midday. At all other times, consumption was residual (Table 7).

**Table 7.** Patterns of alcohol consumption regarding day of the week and meal time among risk drinkers (n = 455).

|  | Weekday | Weekend |
|---|---|---|
| **Wine/Cava** | % | % |
| Breakfast | 3% | 4% |
| Midday | 4% | 7% |
| Lunch | 50% | 65% |
| Afternoon | 0.4% | 0.7% |
| Dinner | 20% | 21% |
| After dinner | 0.2% | 0.2% |
| **Beer/cider** |  |  |
| Breakfast | 2% | 1% |
| Midday | 6% | 8% |
| Lunch | 3% | 7% |
| Afternoon | 3% | 2% |
| Dinner | 3% | 4% |
| After dinner | 0.2% | 0.4% |
| **Liquor** |  |  |
| Breakfast | 2% | 2% |
| Midday | 0.4% | 0.4% |
| Lunch | 2% | 4% |
| Afternoon | 1% | 2% |
| Dinner | 1% | 2% |
| After dinner | 1% | 1% |
| **Fortified wine** |  |  |
| Breakfast | 0.0% | 0.0% |
| Midday | 1% | 3% |
| Lunch | 0.2% | 0.9% |
| Afternoon | 0.0% | 0.7% |
| Dinner | 0.2% | 0.4% |
| After dinner | 0.0% | 0.0% |

Weekdays: Monday to Friday; Weekends: Saturday and Sunday.

### 3.3. Effectiveness of a Brief Intervention on Alcohol Consumption

Only 225 (49%) of the subjects invited to the second phase completed the study (per protocol analysis). Some 25 more, belonging to the intervention group, completed the study but did not perform the complete intervention. They were included in the intention to treat analysis (n = 250). No statistical differences regarding sex, education, or alcohol

consumption were found between those who completed and those who dropped out from the study. Mean age was different at the statistical level ($p = 0.014$) but with still similar ages (70 years who completed, 71 who dropped out).

Comparison groups were comparable regarding sex (60% and 58% men in the control and intervention groups, respectively, $p = 0.769$). Although *p*-values comparing baseline age and SDU were <0.05, differences were little between groups (mean of 70 and 71 years and median of 6 and 7 SDU/week for control and intervention, respectively).

Upon analysis of the data, no significant results were observed when comparing the change in alcohol consumption according to the values of the baseline variables (clinical profile of the subjects). The results of the brief intervention, intention-to-treat analysis (n = 250), are grouped in Table 8. Very similar results were obtained using per protocol analysis (n = 225).

**Table 8.** Distribution and participation by groups and results of the intervention. Intention-to treat analysis.

| | Men | | | | | Women | | | | |
|---|---|---|---|---|---|---|---|---|---|---|
| | Control | | Intervention | | | Control | | Intervention | | |
| | **n** | **%** | **n** | **%** | ***p*** | **n** | **%** | **n** | **%** | ***p*** |
| Recruitment | 127 | | 141 | | | 86 | | 101 | | |
| Final visit | 88 | 69% | 71 | 50% | 0.002 | 49 | 57% | 42 | 42% | 0.036 |
| **Outcome** | | | | | 0.436 | | | | | 0.001 |
| Failure | 41 | 47% | 29 | 41% | | 25 | 51% | 10 | 24% | |
| Partial success | 31 | 35% | 32 | 45% | | 7 | 14% | 20 | 48% | |
| Success | 16 | 18% | 10 | 14% | | 17 | 35% | 12 | 29% | |
| **Grouped score** | | | | | 0.487 | | | | | 0.532 |
| Failure/partial success | 72 | 82% | 61 | 86% | | 32 | 65% | 30 | 71% | |
| Success | 16 | 18% | 10 | 14% | | 17 | 35% | 12 | 29% | |
| **Grouped score** | | | | | 0.468 | | | | | 0.008 |
| Failure | 41 | 47% | 29 | 41% | | 25 | 51% | 10 | 24% | |
| Success/partial success | 47 | 53% | 42 | 59% | | 24 | 49% | 32 | 76% | |
| **Detailed result** | | | | | 0.140 | | | | | 0.018 |
| Increases consumption | 23 | 26% | 16 | 23% | | 14 | 29% | 5 | 12% | |
| Maintains sporadic consumption + medication | 3 | 3% | 0 | 0% | | 2 | 4% | 1 | 2% | |
| Maintains consumption low + medication | 6 | 7% | 10 | 14% | | 3 | 6% | 2 | 5% | |
| Maintains high consumption on one occasion | 0 | 0% | 1 | 1% | | 4 | 8% | 1 | 2% | |
| Maintains high consumption | 9 | 10% | 2 | 3% | | 2 | 4% | 1 | 2% | |
| Reduces consumption but still at risk | 31 | 35% | 32 | 45% | | 7 | 14% | 20 | 48% | |
| Reduces consumption and is no longer at risk | 3 | 3% | 3 | 4% | | 0 | 0% | 0 | 0% | |
| New abstainer | 13 | 15% | 7 | 10% | | 17 | 35% | 12 | 29% | |
| **SDU reduction/week** (mean/SD) | 2.1 | 7.5 | 3.5 | 8.4 | 0.291 | −0.4 | 4.2 | 2.2 | 4.3 | 0.005 |

*p*: test comparing control/intervention in each sex category. Chi-squared or Fisher's exact test for categorical variables, *t*-test for continuous variables.

The logistic analysis shows that the intervention had a significant effect in reducing alcohol consumption (partial or total success) only in women (OR = 3.6 for intention-to-treat analysis and OR = 3.3 for per protocol) (Table 9). Intervention showed no effect comparing total success to partial success or failure, neither overall nor by sex.

**Table 9.** Effect of the brief intervention on partial or total success in reducing alcohol consumption. Mixed-effects logistic regression.

| Intention to Treat | Overall (n = 250) | | | Men (n = 159) | | | Women (n = 91) | | |
|---|---|---|---|---|---|---|---|---|---|
| | OR | CI95% | *p* | OR | CI95% | *p* | OR | CI95% | *p* |
| Crude effect | 2.1 | 0.9 | 4.7 | 0.068 | 1.4 | 0.6 | 3.7 | 0.454 | 3.6 | 1.2 | 10.8 | 0.024 |
| Adjusted for baseline age and SDUs | 2.0 | 0.9 | 4.3 | 0.087 | 1.3 | 0.5 | 3.1 | 0.590 | 3.6 | 1.1 | 11.3 | 0.031 |
| Per protocol | Overall (n = 225) | | | Men (n = 141) | | | Women (n = 84) | | |
| | OR | CI95% | *p* | OR | CI95% | *p* | OR | CI95% | *p* |
| Crude effect | 1.9 | 0.8 | 4.4 | 0.161 | 1.1 | 0.3 | 3.5 | 0.870 | 3.3 | 1.1 | 10.0 | 0.037 |
| Adjusted for baseline age and SDUs | 1.8 | 0.8 | 4.1 | 0.189 | 1.0 | 0.3 | 3.1 | 0.952 | 3.3 | 1.0 | 10.9 | 0.056 |

## 4. Discussion

Liver cirrhosis represents one of the leading causes of mortality globally and ranks second in terms of years of life lost in Europe. This condition can lead to the development of hepatocellular carcinoma, the most prevalent type of liver cancer. These two diseases together contribute to the death of approximately two million individuals annually worldwide [28]. Although cirrhosis caused by hepatitis C virus has decreased due to new treatments, cirrhosis caused by ALD continues to increase, as alcohol consumption is not halted [29]. In turn, the ageing of the population is a reality that profoundly affects the healthcare system. Expectations in Europe are that the elderly will make up 35% of the total population by 2050. These forecasts lead us to seek preventive activities to improve the health of this collective in order to avoid healthcare overload and delay dependency (estimated at 25% by 2025) [30].

To our knowledge, this is the first study to analyse the prevalence of risky alcohol consumption and the effect of a brief intervention against it in the elderly. The results obtained reveal that the number of risky consumers is very high among the population over 64 years of age, accounting for almost half of the sample analysed. Alcohol consumption decreased in those who received the brief intervention. However, the brief intervention only proved to be significantly effective in women.

Notable strengths of the present study include (1) the large number of subjects recruited, which provides a representative sample of the general population of both sexes; and (2) the population-based design of the study with a completely random selection of participants from the SIAP primary care database, which is even more up-to-date than the population census register.

Upon review of the literature, no studies have been found that assess risky alcohol consumption in the elderly. In fact, the only studies that refer to risky drinking focus mostly on adolescents and, in some cases, on adults [31]. It is worth noting that, although no literature exists on this specific population, the results obtained in terms of prevalence of alcohol consumption are very similar to those found in our study [31]. Along the same line, other national and international authors reveal that consumption in adults is slightly lower but still very high [32]. Thus, to date, no work has been found that has described a pattern of low alcohol consumption in the general population, neither in adolescents nor in adults. These data, together with the conclusive results of our study, seem to indicate that there is a social acceptance of alcohol consumption.

Risky alcohol consumption in our environment appears to be related to the major role and relevance given to alcoholic beverages, especially in the social sphere, which has led to the acceptance of this behavioural pattern. What is even more striking is that, as we have seen in this study, this normalization extends to any type of beverage regardless of its alcohol content (wine, beer, liquor, etc.). All this occurs without taking into account how harmful such continued consumption can be for health over a long period of time [32].

Results show that the tendency of men to be risk drinkers is very high, almost doubling the percentage of women. In addition, the interaction between medication and alcohol is relevant in this age group. In this case, though, we found a low prevalence (24%) of risk drinkers who combine little alcohol with medication, most of them having a risky consumption due to the actual consumption of alcoholic beverages.

Half of the subjects targeted as risky drinkers did not perform any follow-up. However, they were similar to the participants who finished the study. Among those included in the brief intervention group, the majority (69%) of the subjects attended all visits, thus showing good adherence to this type of intervention. Similarly, other authors support the positive effects of brief interventions in patients with risk behaviours [33]. In this regard, brief interventions have been used with similar results of acceptance and good continuation response, although with different patterns. On the one hand, Kaner EF et al. in their study obtained similar positive results in both men and women regarding follow-up and effectiveness [16]. On the other hand, other authors conclude that effectiveness is conditioned by sex, with men showing greater progress after the intervention [17]. This marks a notable difference with the results obtained in our project, which showed that the effectiveness of the brief intervention was mainly determined by the female sex, with the effect being minimal in males. This may be due to the fact that in the other studies in which the effectiveness of brief interventions is determined, these are shorter in time. Our study found that women were more adherent to the intervention, especially at the one-year mark. The percentage of men who drop out of follow-up at this point could be the cause of the lower effectiveness reflected in this group.

On the health outcomes associated with interventions, some studies on other behaviours, such as high caloric intake and sedentary lifestyle, potentially associated with obesity and increased cardiovascular morbidity and mortality, have found that the effect of brief interventions was imperceptible, while that of more extended interventions (8–16 weeks) produced significant changes in the short, medium and long term [33,34]. Still, no conclusive data have been found by other authors on brief interventions in the elderly, and there is even more uncertainty as to the same results in the female sex. In fact, a review conducted by Whitlock for the United States Preventive Services Task Force, including 12 controlled trials, revealed that the long-term health outcomes of brief interventions are inconclusive [35].

In contrast to the good adherence to the intervention we found, we were also able to identify that the point of lowest participation and increased dropout was found at the last visit (after one year). This could explain the fact that the effectiveness of the brief intervention does not extend over the long term, probably because of the high prevalence of dropout. Detailing the data on long-term dropout, we observed that sex was a determining factor in this aspect; while women showed good adherence to the intervention until the end, most men only go as far as the 6-month visit. Precisely this behaviour could explain the statistical significance found for the effectiveness of the intervention in women which was not replicated in men. Thus, these data provide us with findings that will allow us to develop even more effective intervention and prevention strategies that can be extrapolated to the majority of the elderly population, irrespective of sex. Based on the results obtained, in which we found that at the 6-month visit there was a high level of participation with a minimal dropout rate, and on what has been described by different authors who have conducted interventions for the same or similar risk behaviours, for future research in this line, we should consider the possibility of modifying the timing of the intervention, moving towards more frequent brief interventions over a period of 6 months [33–35].

This study has some limitations: (1) For the sample size calculation, the estimated prevalence of risk drinkers in >64 years of age was taken into account and not that of associated morbidity, which is lower. (2) The unit of allocation to the intervention and control groups was the health centre to minimize the bias that could occur between patients assigned to both groups. Even so, after the randomization of the centres, we checked that both groups, control and intervention, were comparable. (3) Given that risky alcohol

consumption is stigmatized, it is possible that when quantifying it, some subjects may tend to minimize it. Studies have shown that with appropriate methods it is possible to obtain accurate information on drinking habits. Therefore, a questionnaire based on items about alcohol consumption was used at the same time as questions about other health behaviours. In addition, subjects were unaware of the existence of the other group. As the possibility of an infra report of alcohol consumption, the amount of risk drinkers we have found could be interpreted as a minimum threshold. Any way, we do not expect that this bias could be different between the intervention groups or by sex. (4) Inter-observer variability was minimized by specific training of the staff who collected the data and who implemented the interventions.

## 5. Conclusions

Our findings reveal that the prevalence of risky alcohol consumption in the elderly population is very high in general, being much higher in men than in women. The consumption of alcohol at mealtimes is normalized in this age group, with the highest consumption most commonly occurring during lunch and dinner. The most frequently consumed alcoholic beverage is wine, which is consumed less often on weekdays and increases at weekends.

Alcohol consumption decreased in subjects who received the brief intervention, but only among women. However, while the intervention was successful in reducing alcohol consumption, it was not sufficient to change the subjects from being at-risk drinkers to being non-risk drinkers.

Future research should focus on determining what types of interventions are most effective in this population subgroup and identifying social contexts and health status indicators that may be related to alcohol consumption among the elderly. This is important in order to (1) develop consumption guidelines that detail how the elderly may maintain relatively stable patterns of moderate alcohol consumption without engaging in risky drinking that might negatively influence their health; (2) provide a protocol for action within the framework of preventive activities and health promotion for the elderly that will be effective and compatible with the usual healthcare practice.

**Author Contributions:** Conceptualization of the project P.T., S.M., C.C., A.P.-C., N.M.-J., G.P. and L.C.; methodology, P.T., S.M., C.C., A.P.-C., N.M.-J.; formal analysis, G.P.; investigation, S.M., C.C., A.P.-C., G.D.-F., P.M.-A. and C.R.-P.; writing—original draft preparation, C.C., P.T., S.M., A.P.-C. and G.D.-F.; writing—review and editing, C.C., A.P.-C., P.T., S.M., G.D.-F., I.R.-R., I.A., G.P., C.R.-P., P.M.-A. and L.C.; supervision, P.T. and C.C.; funding acquisition, S.M. and P.T. All authors have read and agreed to the published version of the manuscript.

**Funding:** The project received a research grant from the Carlos III Institute of Health, Ministry of Economy and Competitiveness (Spain), awarded on the call under the Health Strategy Action 2013–2016, within the National Research Program oriented to Societal Challenges, within the Technical, Scientific and Innovation Research National Plan 2013–2016, with reference PI11/01569, co-funded with European Union ERDF funds (European Regional Development Fund).

**Data Availability Statement:** The data presented in this study are available on request from the corresponding author.

**Acknowledgments:** Researchers Maria Dolores Reina Rodriguez, Ana Altaba Barceló, Gregorio Pizarro Romero, Margarita Roset Salla, Sandra Santamaria Bayes and Jaume Roig Llaveria are gratefully acknowledged for their participation in the PI11/01569 project through which funding was obtained to conduct this study.

**Conflicts of Interest:** The authors declare no conflict of interest. The funders had no role in the design of the study; in the collection, analyses, or interpretation of data; in the writing of the manuscript; or in the decision to publish the results.

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
