# Peer review of "Risky Alcohol Consumption in the Elderly: Screening and Brief Intervention from Primary Care. The ALANE Study, a Randomized Clinical Trial"

_beverages, doi:10.3390/beverages9040100_

Round 1

Reviewer 1 Report

Comments and Suggestions for Authors

The manuscript focuses on risky alcohol consumption (RAC) in persons older than 64 and reports on a randomized clinical trial of an intervention aimed at lowering risky alcohol behavior.  Out of the initial 932 subjects interviewed, 642 were considered to have RAC.  The authors conclude from their results that the brief intervention lowered alcohol consumption but not necessarily to a level below what constitutes RAC.  Unfortunately, the Supplemental Materials were not available for download for consideration in this review and they contain more details of the statistical analyses.   Overall, the manuscript is well-written, but contains some major issues with the statistical analyses that need to be addressed.

Major Concerns:

1.            Lines 199-202 suggest that the intervention was randomized using a cluster randomization scheme in which the clinics were the randomization units, and the subjects were the observational units.  However, the statistical analyses do not reflect this.  The flow diagram of the study, Supplemental Figure 1, may have clarified the study design.  Clarification on the study design is needed in Section 2.4.   Furthermore, if the study was designed as a cluster randomized trial, the analyses need to reflect the design and details provided in Section 2.6.

2.            Please clarify if models were run separately for males and females or if sex was just controlled for in the models.  This is not addressed in Section 2.6, yet many results are presented separately for males and females.  Since the power calculations were not stratified by sex, how was the power affected?

3.            Since the dropout rate was much higher than estimated, were the demographics of those who completed different from those who did not?  Were sensitivity analyses conducted with regards to the missing data?

4.            The percentages presented in Table 5 are inconsistent with the text, lines 360-368, that describe them.  The percentages focus on the percent of abstainers, etc that fall into each demographic category, but the text reads as if the same percentage reflect the percent of the demographic category that are abstainers.

5.            In Table 7, the SDU (standard drinking units) analyses were ran using ANOVA.  However, no details are provided on a review of the appropriateness of using ANOVA, in particular the normality of the data.  With a relatively large proportion of the subjects being abstainers, one would expect the data to be skewed or zero-inflated.

6.            For many of the categorical analyses, the Chi-square would not be appropriate due to greater than 25% of the cells having expected values of less than 5.  Fishers Exact Test should be used.

7.            Lines 523-524 states “Even so, in the randomization of the centers we adjusted for the characteristics of the centers.”  Please explain further and include relevant details in the Methods section.

Other Concerns:

1.            The footnotes on Tables 3 and 4 need clarification (p<0.001 is just listed below.

2.            In Table 4, use the same terms in the table and the text would make the results easier for a reader to follow.

3.            For Table 6, details should be presented on the total N.  Is this table focused on non-abstainers or the full population?

4.            What do the p-values in Table 8 represent for the first section (Recruitment vs Final Visit) and in the 5th section (Analysis by Protocol)?  If this is significant, then details should be provided on any differences seen at baseline between those that completed and those that dropped out?  (See point #3 above.)  Also, was SDU reduction/week normally distributed?

Comments on the Quality of English Language

Minor English editing is needed.  The few grammatical errors that caught my attention may simply be typos and were not distracting as a whole.

Author Response

We fully agree with your requests and comments. Please, find below, point by point, our answers.

Reviewer 2 Report

Comments and Suggestions for Authors

The manuscript under review discusses the prevalence of risky alcohol consumption in elderly populations and evaluates the effectiveness of a brief intervention to reduce alcohol consumption among them. The authors provide valuable insights into the health risks associated with alcohol consumption and the potential benefits of interventions targeting this issue.

The introduction provides a comprehensive overview of the significance of risky alcohol consumption, especially among the elderly, in the context of global health. The problem statement is well-structured and highlights the urgency of addressing this issue.

Recommendation: However, there is a need to clarify the aim and methods of the study more explicitly. The introduction could benefit from a clear and concise statement of the research objectives and the hypothesis being tested.

Sample Size and Participant Withdrawal: One major concern is the low sample size and the high rate of participant withdrawal, which could significantly impact the study's significance and applicability. The authors should address these issues in the manuscript to explain their potential impact on the results and conclusions. Additionally, suggestions for improving participant retention and data collection should be discussed.

Tables: The manuscript contains extensive tables with a large amount of data. While data presentation is essential, the tables need revision to improve clarity. Consider using subheadings, column/row headings, and possibly visual aids to facilitate better understanding for the readers.

Bias in Alcohol Consumption Evaluation: The manuscript mentions potential biases in the evaluation of alcohol consumption, specifically regarding the stigma associated with risky alcohol consumption. This is an important issue, but it needs further elaboration and exploration in the manuscript. Discussing methods used to minimize or account for this bias would enhance the research's validity.

Analysis of Social Factors: The manuscript lacks an analysis of social factors that could influence participant withdrawal. Understanding the sociocultural, economic, or psychological factors contributing to participant attrition can provide valuable insights and strengthen the study's conclusions.

Conclusion: In conclusion, the manuscript provides valuable insights into the prevalence of risky alcohol consumption among the elderly and the potential effectiveness of brief interventions. However, the manuscript needs revisions to clarify its aim, address issues related to sample size and participant withdrawal, and improve the presentation of tables. Additionally, exploring the impact of social factors and biases in alcohol consumption assessment would enhance the study's quality. With these improvements, the manuscript could make a valuable contribution to the field of public health.

Comments on the Quality of English Language

Minor editing of English language required

Author Response

(The authors gave the same response as above.)

Round 2

Reviewer 1 Report

Comments and Suggestions for Authors

Thank you for your clear and detailed response to the first review.  Your changes have answered all of my initial questions and concerns.  In particular, the statistical methods section provides the needed details of the analyses and illustrates that the cluster randomization was handled properly.

Three minor corrections to be noted: 

Table 4 - Regular non-risk consumption risk - the last 7 needs a "%" for consistency.

Lines 357-358:  Adjust sentence to reflect that >75 years had the lowest prevalence among age groups.  Currently, it reads as if it is lowest for any sub-group.

Table 5 - Age (Mean SD) - the last column needs the "%" removed.

Author Response

Dear reviewer, many thanks for you helpful comments. Please, find enclosed the corrected manuscript.

Reviewer 2 Report

Comments and Suggestions for Authors

the manuscript significantly improved after the requested revisions.

Comments on the Quality of English Language

after minor changes and language check is is suitable to pr